# The Role of Telehealth to Assist In-Home tDCS: Opportunities, Promising Results and Acceptability

**DOI:** 10.3390/brainsci8060102

**Published:** 2018-06-07

**Authors:** Brenton Hordacre

**Affiliations:** The Sansom Institute for Health Research, School of Health Sciences, University of South Australia, Adelaide 5001, Australia; Brenton.hordacre@unisa.edu.au; Tel.: +61-83021286

**Keywords:** transcranial direct current stimulation, remotely supervised, telehealth, telemedicine, telerehabilitation

## Abstract

Transcranial direct current stimulation (tDCS) has shown great promise as a neuromodulatory intervention capable of improving behavioral outcomes in a range of neurological and psychiatric populations. Evidence indicates that the neuromodulatory effect of stimulation may be cumulative, with greater improvements in behavior observed following multiple treatment sessions. However, the requirement to attend clinical or research departments for multiple treatment sessions may present a barrier for many people, particularly those with greater disability or living remotely. The portability of tDCS suggests that in-home stimulation may become an avenue for further investigation. However, safe and effective use of tDCS by a participant within their home requires a form of monitoring. This review discusses how telehealth may provide real-time visual monitoring to ensure correct tDCS set-up and adherence to stimulation protocols, manage technical issues and monitor adverse events. The combination of telehealth to supplement in-home tDCS use has potential to transform the way tDCS is delivered.

## 1. Introduction

Interventions capable of modulating cortical function have significant potential to be utilized in a therapeutic manner. This may be particularly true for neurological and psychiatric conditions, such as stroke, multiple sclerosis and depression, which are underpinned by abnormalities in brain function [1,2,3,4,5]. In these populations, interventions which target the brain have shown potential to help improve behavioral or psychological outcomes. For example, brain stimulation has shown promising results in both stroke and treatment-resistant depression to reduce impairment and improve mood respectively [6,7,8]. As a result, brain stimulation may be a valuable tool to treat various neurological and psychiatric conditions.

One form of non-invasive brain stimulation is transcranial direct current stimulation (tDCS). TDCS involves weak (0.5–2 mA) direct current, passing between two, or more, electrodes positioned over the scalp. Like other forms on non-invasive brain stimulation, tDCS is capable of selectively increasing or decreasing cortical excitability. For anodal tDCS, where the anode is placed on the scalp overlying the target cortical region, current causes a depolarization of the neuronal cell membrane, increasing excitability and allowing spontaneous cell firing [9]. The increase in excitability is observed beyond the period of stimulation, with pharmacology studies demonstrating that this neuroplastic effect is mediated by mechanisms which resemble long-term potentiation-like synaptic plasticity [10]. For cathodal tDCS, the cathode is positioned on the scalp overlying the cortical target, causing a hyperpolarization of the neuronal cell membrane, decreasing excitability and reducing spontaneous cell firing [9]. This decrease in excitability is observed beyond the period of stimulation, and likely underpinned mechanisms resembling long-term depression-like synaptic plasticity [10]. 

Evidence suggests that responses to tDCS may be cumulative with stronger responses observed following multiple treatment sessions [11]. Therefore, repeated stimulation sessions will likely be a hallmark of future tDCS use in clinical and research environments. In support, many studies have applied tDCS over a number of consecutive sessions, spanning days, months or even years [6,12,13]. The requirement for participants to attend research departments for multiple experimental sessions, or clinical facilities for repeated treatment sessions, can be challenging and costly for some individuals, potentially requiring transportation to the clinic and incurring clinic charges to receive treatment. Therefore, the need to attend multiple stimulation sessions may present as a barrier for participation in research studies or for receiving clinical services. Indeed, many interventional studies using tDCS over multiple sessions are relatively small in size with most having 30 or less participants in the active stimulation arm of the trial [14,15,16]. However, one advantage of tDCS over other forms of non-invasive brain stimulation is that the device is easily portable. This opens up the possibility that tDCS treatment sessions could be delivered in a participant’s home, reducing travel burden associated with attending consecutive stimulation sessions.

While tDCS is portable, suggesting potential for in-home use, its application should remain monitored by a trained professional. Incorrect use can not only limit possibility of beneficial effects of stimulation being observed, but may contribute to the occurrence of serious adverse events. For example, evidence from animal models suggests brain injury resulting from direct currents can occur at high intensities [17,18], several orders of magnitude greater than those currently used in human research. Conversely, stimulation delivered for very brief periods has been shown to produce no lasting changes in cortical excitability. Evidence for this is observed in common sham stimulation protocols, where a brief period (~30 s) of current is delivered to mimic the initial cutaneous sensation of tDCS, but appears to be insufficient to change cortical excitability [9,19]. Correct and safe use of tDCS requires knowledge of current intensity, stimulation duration and current density. While several home or clinic based stimulators are now available which can be pre-programmed by an experienced technician, preventing users from modifying stimulator parameters, electrode positioning and preparation remain vital aspects of safe and effective tDCS usage. Poor electrode preparation and skin contact can lead to skin irritation or skin lesions [20,21,22,23]. High impedance may prevent stimulation from starting or interrupt current during a stimulation protocol. Incorrect electrode positioning could result in current not being delivered to the intended target in the cortex. At best, this may result in no behavioral changes being observed, while at worst, delivering current to an unintended cortical region may result in detrimental behavioral effects. Even well trained participants are likely to require some form of monitoring by trained tDCS technicians for correct home stimulation.

The purpose of this review is to describe how the use of information and communication technologies may serve as a tool to facilitate and monitor in-home tDCS. This review will specifically focus on real-time visual monitoring for patient groups likely to be candidates for tDCS usage in research and clinical domains. First, this review will highlight challenges faced by un-monitored in-home tDCS. Following this, promising results from current studies using real-time visual monitoring to facilitate in-home tDCS will be discussed. This review will also identify patient populations that are likely to be appropriate for using information and communication technologies and discuss the preferences and acceptability of using this new technology. Finally, this review will discuss limitations of this approach and directions for future research. Identifying approaches to achieve safe and effective use of in-home tDCS is critical to facilitate uptake of this potentially beneficial neuromodulatory technique in the home. 

## 2. Challenges Facing In-Home tDCS

Few studies have trialed in-home use of tDCS with various patient groups to determine feasibility of this approach. However, several technical issues have been identified. For example, Hyvärinen et al. investigated the use of tDCS for treatment of tinnitus in a double-blind sham-controlled study [24]. The first treatment was completed in the clinic after a training session, followed by nine daily sessions in each participant’s own home. A custom-made tDCS electrode cap which had sewn-on electrode positioning guides combined with a pre-programmed tDCS device was used to ensure ease of application and correct usage. Patients were instructed to keep a treatment diary to record time of each session and difficulties with preparation or stimulation. During the period of un-monitored in-home tDCS, one participant experienced a slight skin burn associated with the stimulation. It was thought this may have occurred due to incorrect placement of the cathode electrode with the wrong side of electrode in contact with the scalp. The authors acknowledge that this may have resulted in a higher current density under the terminal wire. 

In a case study which investigated maintenance tDCS in a patient with schizophrenia, a medically qualified family member was trained to deliver stimulation up to two times daily in the participant’s home [12]. Home stimulation was delivered over a period of three years, with the authors reporting two periods of marked increase in symptoms which resulted in the patient returning to near baseline levels of severity. These periods were associated with different technical issues which included the family unknowingly interchanging the anode and cathode following repair to electrode leads, and incorrect electrode placement which resulted in current delivered to unintended cortical regions.

Similarly, technical issues were reported by Hagenacker et al. who investigated the use of tDCS for treating trigeminal neuralgia in a randomized cross-over study [25]. Participants and a family member were instructed on the use of tDCS and then subsequently performed stimulation daily at home for a period of 14 days. To ensure the correct application of tDCS, the stimulator recorded each application, and participants were able to call the study team if there were malfunctions or handling problems. Of the seventeen patients enrolled in the study, one did not appear to stimulate at all as recorded by the stimulator log. It may have been that this patient was unaware that they were not correctly starting the stimulator at each session.

These technical issues, which resulted in incorrect, or non-use, of in-home tDCS and some minor adverse events, could have been avoided or appropriately managed with real-time monitoring by medical or research technicians. Using a real-time visual monitoring process may allow confirmation of correct stimulator set-up, electrode positioning and ensuring stimulation starts and is completed as planned. Therefore, there is a need to remotely monitor use of in-home tDCS in real-time to improve treatment compliance, ensure safe tDCS usage and achieve best outcomes from the stimulation intervention. 

## 3. Opportunities: How Monitoring Could Be Achieved 

One approach to achieve real-time monitoring of in-home tDCS is to use a domiciliary care model where a technician would attend the participant’s home for each tDCS session and ensure correct application. This approach has successfully been used in the past and reduced travel burden for study participants [26]. However, domiciliary approaches require greater time commitments and travel from research staff in order to attend the participant’s home for each stimulation session. In addition, this approach does not facilitate clinical or research studies to undertake stimulation for people that live remotely as travel time by the technician to attend the participant’s home would be too great. An alternative approach would be to utilize advances in information and communication technologies to provide remote, real-time, visual and auditory confirmation of correct and safe tDCS application. Videoconferencing can occur through various platforms, including computers, tablets or mobile phone devices (Figure 1). With the use of these technologies, it may be possible for technicians or support staff to observe electrode placement on the scalp, and ensure stimulation is delivered within the required parameters. Where required, feedback could be provided to correct issues and facilitate safe and effective use. The use of information and communication technology in health service delivery, known as telehealth, could have enormous potential to help deliver tDCS remotely, which provides opportunities to broaden research and therapeutic use of this neuromodulatory technique.

### Telehealth Explained

Telehealth utilizes advanced telecommunication technologies to deliver health services remotely under supervision. Information and communication technologies can enable visual and auditory communication in real time between the healthcare professional and the patient in a remote location. Services provided by telehealth can include evaluation, assessment, monitoring, prevention, intervention, supervision, education, consultation and coaching [27]. Using technology in this manner can break down barriers imposed by face-to-face services, and may be an appropriate strategy to increase accessibility for those with disability or who live remotely to the clinic or hospital [28,29,30]. While requirement for technology use may be confronting for elderly patients or those with limited experience in using technology, these characteristics do not appear to impede utility of telehealth [31]. Indeed, many participants report high levels of satisfaction with the use of telehealth technologies [31]. Previous studies using various forms of telehealth technology demonstrate successful use in a range of clinical scenarios, including delivery of services to people suffering from depression [32] and stroke [33], as well as elderly caregivers [34] and a platform for delivering psychological interventions [35]. Furthermore, a recent systematic review reported several positive health outcomes associated with telehealth, including improved quality of life, clinical outcomes and health status [36]. Telehealth has also been associated with reduced hospital re-admissions, reduced mortality and increased service and patient contact time [31,37]. While telehealth is unlikely to replace all face-to-face consultations, it may be used to supplement these appointments to reduce travel burden, waiting time, or cost and time associated with clinical or research staff travelling to participants’ houses. 

Although approaches to best utilize this technology are still debated and the focus of ongoing research [38], one novel approach may be to use telehealth as a monitoring and coaching strategy for in-home tDCS applications. The availability of less resource-intensive approaches to implement tDCS experimental trials or treatment protocols which can span a number of repeated consecutive daily sessions is important given the mobility challenges faced by those requiring this therapy. The implementation of this approach to monitor in-home tDCS could be achieved using off-the-shelf-technologies, such as laptops, tablets and mobile phones that are loaded with videoconferencing software [31]. Indeed, it is likely that a proportion of patients or study participants would already possess the required hardware to facilitate real-time monitoring of in-home tDCS using telehealth. Using a telehealth approach to monitor in-home tDCS use may be an appropriate strategy to improve compliance with the stimulation protocol, manage technical issues and monitor adverse effects associated with stimulation. 

## 4. Promising Results: In-Home tDCS Monitored by Telehealth 

Several studies have demonstrated the potential for telehealth approaches to facilitate in-home use of tDCS in various neurological and psychiatric patient populations. For example, a series of recent studies describing a carefully designed remote tDCS supervision protocol have been conducted with people who have multiple sclerosis [39,40,41]. Studies applied tDCS in the participant’s home to either test feasibility of in-home use [41], improve symptoms of fatigue [39] or increase benefits of cognitive training [40]. Stimulation was delivered using a Soterix mini-Clinical Trials tDCS device (Soterix Medical, New York, NY, USA) at 1.5–2 mA over a period of 10 to 20 sessions. To monitor in-home tDCS use, the research team employed several strategies, including use of telehealth technologies to facilitate remote supervision. Participants were provided with laptop computers as part of the in-home stimulation kit. Laptops were configured with videoconferencing software, which allowed research staff to confirm correct electrode placement and provide a single use code that unlocked the tDCS device for one stimulation session. The session would then be conducted under supervision with the research staff monitoring common side effects of tDCS. Although minor adverse events occurred, the authors reported excellent levels of protocol compliance without the occurrence of technical issues, such as incorrect electrode placement or stimulations not being delivered.

In another study, the use of in-home tDCS was investigated for people with Mal de Debarquement Syndrome (MdDS) [42]. In this population in-home tDCS is a particularly appropriate strategy as MdDS is characterized by dizziness which is exacerbated by travel or motion, making it difficult for patients to travel for daily treatment sessions in a clinic. After receiving training in the use of the in-home tDCS device, patients self-administered 20 sessions of tDCS over a period of 4 week within their own home. The investigators used several telehealth-based strategies to confirm correct tDCS set-up, monitor protocol compliance and record adverse events. These strategies included confirmation of correct electrode positioning using a webcam to either videoconference with the research team or to send pictures to the research team. Online tracking via a personalized web link was used to monitor protocol compliance and symptoms. Participants were required to log onto the webpage as a daily check-in. If this was not completed as per the protocol, a member of the research team would contact the participant via phone or email. The authors report high compliance and excellent safety with monitored in-home tDCS. There were no instances of skin irritation or skin burns, which are common adverse events to tDCS. 

Telehealth approaches to monitor use of tDCS have also been reported in studies using tDCS for maintenance of managing symptoms of various diseases over extended periods of time. In a case-study of a male with macrophagic myofasciitis who lived far (approximately 300 km) from the clinic, pain was effectively managed using tDCS delivered to both motor cortices [43]. A Newronika tDCS device (Newronika, Milan, Italy) was provided to the patient along with a webcam to allow a ‘Tele-NIBS’ approach. Prior to beginning the weekly maintenance session, Skype was used to confirm electrodes were placed correctly, the cables were connected appropriately and that stimulation started as programmed. The participant was monitored visually for the duration of stimulation (20 min) and for a few minutes after the end of stimulation. 

Home based tDCS, monitored using telehealth, has also been investigated among people with depression. In an on-going study, ten participants self-administered tDCS, delivered at 2 mA for 30 min per session over 20 consecutive weekdays. Monitoring was provided via a video link across the initial sessions, followed by daily phone or email contact [44]. The authors report excellent levels of compliance with no significant adverse events.

Although there are a few studies which have implemented a telehealth approach to monitor in-home tDCS, it appears to help avoid technical issues and facilitate compliance with stimulation protocols. This would suggest that the provision of real-time visual monitoring via a telehealth approach is a key aspect of future in-home tDCS research or clinical services.

## 5. Acceptability: Who Is Appropriate for Telehealth and What Are Their Preferences

### 5.1. Populations Suitable for Telehealth

Currently, telehealth has been used to monitor in-home tDCS use among people with multiple sclerosis [39,40,41], Parkinson’s disease [45], MdDS [42], pain [43] and depression [44], suggesting potential use across a range of patient groups. Beyond monitoring of in-home tDCS, telehealth has been implemented in a broad range of patient populations [46,47,48,49,50,51,52], suggesting enormous potential to achieve real-time visual monitoring for people who are likely to benefit from tDCS. Although both age and familiarity with technology do not appear to be barriers to telehealth or influence the acceptability of this service, familiarity with technology correlates with therapy dosage [31]. Those who were more familiar with technology were found to have a greater number of videoconferences and have more time spent on videoconferencing [31]. The authors suggest that adequate training needs to be considered to enable successful participation. An approach to reinforce the required training might include educating both the patient or study participant along with a supportive family member. Previous studies using in-home tDCS have used this approach so that the support person can assist with application of tDCS [39,40,41], particularly for those study participants with high levels of disability. While disability level may not restrict use of telehealth technology, having an additional support person trained to assist could facilitate uptake and acceptance of this approach.

### 5.2. User Preferences

Understanding experiences and perceptions regarding use of telehealth modalities will be crucial for successful implementation of in-home tDCS. In a recent study, the authors sought to determine the views and preferences of older adults regarding basic features of telehealth used to provide a service [53]. Using a discrete choice experiment, the authors reported that patients were accepting of telehealth services which covered all aspects of care, were relatively inexpensive and available for people living in remote regions. Participants also preferred telehealth to be provided by clinicians who were positive about telehealth and suggested that the service should be offered to people with some prior technology experience. This preference for telehealth to be provided to people with prior technology experience may imply that the technology platform (laptop, tablet, and mobile phone) is tailored toward the individuals experience and preference. For example, if a participant was comfortable with use of a laptop, it may be preferable to utilize this platform to perform videoconferencing. Importantly, participants did not feel telehealth led to a loss of privacy and confidentiality. In a qualitative study which obtained the perspectives of community dwelling participants who had received an in-home telehealth program comprising use of iPads for videoconferencing, there were several major themes which emerged [54]. These key themes were that telehealth should be convenient, promote motivation and self-awareness, foster positive therapeutic relationships, and does not replace face-to-face rehabilitation therapies. Together these studies indicate telehealth is an acceptable approach and may be suitable to monitor in-home tDCS use if it can be achieved in an inexpensive manner, is convenient and tailored toward prior technology experience. Key points regarding use of telehealth to monitor in-home tDCS are summarized in Box 1.

Box 1Key points regarding the use of telehealth to monitor in-home tDCS.*Monitoring In-Home tDCS Use with Telehealth* 
Telehealth can help avoid technical issues and limit
adverse eventsTelehealth appears suitable for many patient groupsTelehealth should be inexpensive, convenient and tailored
toward prior technology experienceUsing a telehealth approach to monitor in-home tDCS should
be combined with appropriate training People with high impairment or poor cognitive function may
benefit from assistance of a family member or a support person


## 6. Limitations of Using Telehealth to Monitor In-Home tDCS

While telehealth may be a promising approach to facilitate in-home tDCS use, there are several limitations which should be considered prior to utilizing this technology. Considering technical aspects first, information and communication technologies used for videoconferencing require access to reliable and stable internet services. While available networks continue to expand and improve in quality and speed, those living in extremely remote regions may have limited access to acceptable Internet services. The costs associated with implementing telehealth may also be a barrier for both research and clinical services. While various hardware platforms are available to facilitate videoconferencing (e.g. mobile phone, tablet or laptop), some of which may already be owned by the patient or the study participant, these may have to be purchased and regularly updated by the clinical or research team. It should also be highlighted that telehealth technology is not able to limit or monitor delivery of tDCS. As a result, it may be possible for tDCS to be delivered too frequently, which has potential to cause harm to the patient. However, depending on the tDCS device used, strategies may be available to ensure delivery is limited to the intended level of use.

At the patient or study participant level, telehealth technology to monitor in-home tDCS use may be intimidating or confronting to those who are not familiar with the technology. While this has previously been shown not to prevent the use of telehealth, there appears to be greater utilization in those familiar with technology [31]. Questionnaires, such as the modified computer self-efficacy scale, have been validated for use with older adults or people with disability to help identify people who may be more open to use of new technology [55]. It may be that those who are less familiar with technology are still capable of partaking in telehealth, but require more comprehensive training. 

## 7. Future Directions

Telehealth has potential to provide a monitoring approach for in-home tDCS use (Box 1). However, to date, only few studies have investigated the feasibility of this approach [39,40,41,42,43,44]. While results are positive, larger trials are required to gain further perspectives around protocol adherence and safety using this approach. To further support this, future studies should identify appropriate methodologies to monitor symptoms and track progress during the course of tDCS treatment. Future studies would also be required to determine which patient groups are not suitable for using telehealth. While it is not clear whether those with high physical or cognitive impairment may find it difficult to use telehealth technologies, it may be that a family member or a support person can assist within the home environment. 

## 8. Conclusions

A telehealth approach to provide real-time visual monitoring of in-home tDCS use has significant potential to facilitate stimulation within an individual’s home. With the assistance of technology to provide videoconferencing capabilities, technicians can monitor in-home tDCS set-up, delivery of stimulation, compliance with the stimulation protocol and monitor issues that arise. As in-home tDCS devices continue to evolve, it is likely that telehealth approaches will become a cornerstone to provide monitoring of safe and effective tDCS application. Future studies considering in-home tDCS should consider a monitoring approach such as that provided by telehealth.

## Figures and Tables

**Figure 1 brainsci-08-00102-f001:**
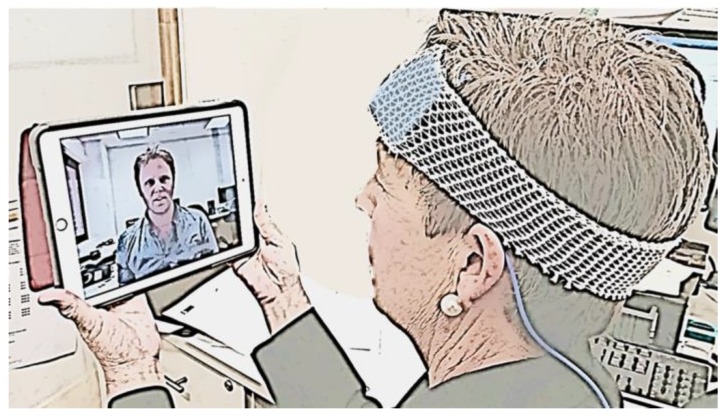
A participant preparing for in-home tDCS confirms correct tDCS electrode set-up and has stimulation monitored in real time via a videoconference with a study technician.

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
