# Peer review of "The Role of Telehealth to Assist In-Home tDCS: Opportunities, Promising Results and Acceptability"

_brainsci, 2018, doi:10.3390/brainsci8060102_

Round 1

Reviewer 1 Report

The author provides a review on in-home tDCS and telehealth approaches to support tDCS in disabled persons.

It is fluently written and well readable, but the main message is unclear. It is written as a review but it is more a position or methods paper. For a review paper, it does not cover all relevant literature (see Palm et al., Neuromodulation 2017, for a review of at-home tDCS and disambiguation of different terms). For a position paper on telehealth, it does not provide enough details or proposals for driving this approach (here, several publications by the Charvet and Kasschau research group already defined in detail how to perform remotely supervised tDCS).

It is suggested that the author largely rewrites this manuscript to clearly define the aim of this work. Otherwise it is an incomplete narrative review without relevant contribution to the advancement of this field of research.

Line 31: Sentence is unclear. Please correct.

Author Response

A word document with responses has been uploaded

Reviewer 2 Report

Thank you for inviting me to review the paper “The role of Telehealth to assist In-home tDCS: Opportunities, Promising Results and Acceptability”. This is an excellent comprehensive review of current status of tDCS and home treatment is the hot topic. I recommend publication in Brain Sciences with minor revision.

1.     In line 31, please specify the word, “non-resistant” depression to improve mood (reference PMID:29763711)

2.     In line 51, please indicate that one of the reasons for limiting repeating treatment session is high cost involving clinic charge and transportation.

3.     In line 150, the authors should mention specific benefits of telehealth related to clinical indications for tdcs. Please add the statement that telehealth was applied to treat patients suffering from depression (Reference: PMID: 29138195) and stroke (Reference: PMID: 27061386) as well as supporting elderly caregivers (Reference: PMID: 27129032) and delivering psychological intervention (Reference: PMID: 26409514).

4.     Under Section 6 limitations, please state that one important limitation, the current telehealth cannot stop a person who administers tDCS too frequently, which may cause harm to the patient.

5.     Under Section 7 future directions, please state that future telehealth will include questionnaires to monitor symptoms and track progress during the course of tDCS treatment.

Author Response

(The authors gave the same response as above.)

Round 2

Reviewer 1 Report

The author significantly improved the manucript to give a clear aim and pupose.